# Succinylation of Polyallylamine: Influence on Biological Efficacy and the Formation of Electrospun Fibers

**DOI:** 10.3390/polym13172840

**Published:** 2021-08-24

**Authors:** Lucija Jurko, Matej Bračič, Silvo Hribernik, Damjan Makuc, Janez Plavec, Filip Jerenec, Sonja Žabkar, Nenad Gubeljak, Alja Štern, Rupert Kargl

**Affiliations:** 1Laboratory for Characterisation and Processing of Polymers, Faculty of Mechanical Engineering, University of Maribor, Smetanova 17, 2000 Maribor, Slovenia; lucija.jurko@um.si (L.J.); matej.bracic@um.si (M.B.); silvo.hribernik@um.si (S.H.); 2Faculty of Electrical Engineering and Computer Science, University of Maribor, Smetanova 17, 2000 Maribor, Slovenia; 3Slovenian NMR Center, National Institute of Chemistry, Hajdrihova 19, 1000 Ljubljana, Slovenia; damjan.makuc@ki.si (D.M.); janez.plavec@ki.si (J.P.); 4EN-FIST Centre of Excellence, Trg Osvobodilne Fronte 13, 1000 Ljubljana, Slovenia; 5Faculty of Chemistry and Chemical Technology, University of Ljubljana, Večna Pot 113, 1000 Ljubljana, Slovenia; 6Faculty of Mechanical Engineering, University of Maribor, Smetanova 17, 2000 Maribor, Slovenia; filip.jerenec@um.si (F.J.); nenad.gubeljak@um.si (N.G.); 7Department of Genetic Toxicology and Cancer Biology, National Institute of Biology, Večna Pot 11, 1000 Ljubljana, Slovenia; sonja.zabkar@nib.si (S.Ž.); alja.stern@nib.si (A.Š.); 8Institute for Chemistry and Technology of Biobased System (IBioSys), Graz University of Technology, Stremayrgasse 9, 8010 Graz, Steiermark, Austria

**Keywords:** polyallylamine hydrochloride, succinylation, aqueous chemistry, cytotoxicity, antimicrobial effect, electrospinning, nanofibers: mouse L929 fibroblasts, *Staphylococcus aureus*, *Pseudomonas aeruginosa*

## Abstract

Succinylation of proteins is a commonly encountered reaction in biology and introduces negatively charged carboxylates on previously basic primary amine groups of amino acid residues. In analogy, this work investigates the succinylation of primary amines of the synthetic polyelectrolyte polyallylamine (PAA). It investigates the influence of the degree of succinylation on the cytotoxicity and antibacterial activity of the resulting polymers. Succinylation was performed in water with varying amounts of succinic anhydride and at different pH values. The PAA derivatives were analyzed in detail with respect to molecular structure using nuclear magnetic resonance and infrared absorbance spectroscopy. Polyelectrolyte and potentiometric charge titrations were used to elucidate charge ratios between primary amines and carboxylates in the polymers. The obtained materials were then evaluated with respect to their minimum inhibitory concentration against *Staphylococcus aureus* and *Pseudomonas aeruginosa*. The biocompatibility was assessed using mouse L929 fibroblasts. The degree of succinylation decreased cytotoxicity but more significantly reduced antibacterial efficacy, demonstrating the sensitivity of the fibroblast cells against this type of ampholytic polyelectrolytes. The obtained polymers were finally electrospun into microfiber webs in combination with neutral water-soluble polyvinyl alcohol. The resulting non-woven could have the potential to be used as wound dressing materials or coatings.

## 1. Introduction

Polyallylamine hydrochloride (PAA) is a synthetic polyelectrolyte obtained by polymerization of allylamine. It has been previously used for the production of nanoparticles [1], for polyelectrolyte multilayers [2], for purification of water [3], as an ion exchanger [4], or for the removal or absorbance of heavy and rare metals [5]. Polyallylamine is also known to have antimicrobial properties [6]. These properties are explained by their positive charge in the protonated form, which allows the polymer to ionically bind and insert into parts of negatively charged bacterial cell membranes, causing disruption and death of the organism [7,8]. Interest persists in finding suitable antimicrobial agents against the growth of *Pseudomonas aeruginosa* and *Staphylococcus aureus* which are frequently found on human skin but can become potentially pathogenic [9]. Cationic moieties are known excellent non-selective antimicrobial agents, but can also exert undesired cytotoxic activity on mammalian cells, including fibroblasts [10,11]. Several attempts have therefore been made to reduce cytotoxicity by derivatization of polyamines. These included N-acylation [6], grafting of guanidine derivatives [8], or cross-linking [12]. Reactions like methylation, acetylation, or addition of longer alkyl chains were usually conducted in organic solvents to decrease the rate of hydrolysis as a competing reaction [13]. Succinylation of polyamines caused a reduction of cytotoxicity [14,15] and is also often encountered naturally in proteins with plenty of biological functions [16]. Succinylated gelatine is for instance commercially used as blood plasma volume expander [17]. Although there were some succinylation reactions conducted in THF [18], DMSO [19], and DMF [20], reactions in aqueous solutions are of greater interest, for biomedical applications and due to environmental issues. Succinylated materials could also be used for the production of antimicrobial wound dressings [21]. To the best of our knowledge, no reports exist on the electrospinning of succinylated PAA, even though this method is often investigated to produce wound dressing materials. In this work, the succinylation of PAA and its influence on the biocompatibility in a mouse fibroblast model system (L929 cells) and the antimicrobial efficacy against *S. aureus* and *P. aeruginosa* were investigated. Reactions between PAA and succinic anhydride were performed at different molar ratios in alkaline water, avoiding organic solvents. NMR and ATR-IR spectroscopy were used together with potentiometric and polyelectrolyte charge titrations, to confirm covalent derivatization and charge ratios of the obtained polyampholytes. In combination with polyvinyl alcohol, the materials were electrospun into non-wovens, with the intention to be used as wound dressing materials. The influence of the spinning parameters and polyampholyte on the fiber diameter was assessed.

## 2. Materials and Methods

### 2.1. Materials

Polyallylamine hydrochloride (PAA; Mw: 50,000 g mol^−1^), aqueous sodium hydroxide (NaOH; 1 M), polyvinyl alcohol (PVA; Mw: 89,000–98,000 g mol^−1^; 99+% hydrolyzed); Toluidine Blue (certified) and polystyrene sulfonic acid, sodium salt (PSS); sodium pyruvate (P2256); L-glutamine (G7513), penicillin/streptomycin (1%, P0781), dimethylsulphoxide (DMSO) were purchased from Sigma-Aldrich (St. Louis, MO, USA). Succinic anhydride and potassium hydroxide (KOH; pellets) were purchased from Merck (Hohenbrunn, Germany). Hydrochloric acid (HCl; 0.1 M) was purchased from Kefo (Ljubljana, Slovenia). Dialysis tubes (regenerated cellulose membrane, MWCO 6–8 kDa) were purchased from Carl Roth (Karlsruhe, Germany). Minimum Essential Medium (MEM; 51200-046) and Foetal bovine serum (100500-064) were from Gibco™ (Amarillo, TX, USA). Trypsin, was from Invitrogen (Waltham, MA, USA). Etoposide (ET) was from Santa Cruz Biotechnology (Dallas, TX, USA). 3-(4,5-dimethylthiazol-2-yl)-2,5-diphenyltetrazolium bromide (MTT) was from Abcam (Cambridge, UK). Phosphate buffered saline (PBS) was from PAA Laboratories (Toronto, Canada). Ultra-pure water from a Millipore (MA, USA) water purification system (resistivity ≥ 18.2 MΩ cm, pH 6.8) was used for the preparation of all aqueous solutions throughout the whole work.

### 2.2. Succinylation of Polyallylamine

Polyallylamine hydrochloride was dissolved in deionized water under shaking (10% (*w*/*w*), 25 °C, 200 rpm), and the pH adjusted to 9 for deprotonation of most of the amine groups. With the complete dissolution of the polymer, and adjustment of the pH, different molar ratios of succinic anhydride were gradually added to the solution. After readjustment of pH to 9 < pH < 11, the homogeneous solution was left to react for 24 h under ambient conditions. The product was dialyzed for 4 days against ultra-pure water, and later lyophilized (−35 °C; ~2.0 × 10^−3^ mbar, 4 days) to obtain the final powdered product.

Amidation of PAA (Figure 1) with succinic anhydride (SA) can occur simultaneously to anhydride hydrolysis. In this work, the materials obtained are labelled as PAA-xSA with x representing the molar ratio of succinic anhydride to the amine group of PAA. Together with unmodified PAA, five materials were produced labelled PAA-0.5SA; PAA-1SA; PAA-2.5SA; PAA-5SA; and PAA-10SA.

### 2.3. Attenuated Total Infrared Reflectance Infrared Spectroscopy (ATR-IR)

Attenuated total reflection infrared spectroscopy (ATR-IR) spectra of all samples were measured using a PerkinElmer FTIR System Spectrum GX Series-73565 at a scan range of 4000–650 cm^−1^. A total of 32 scans were performed at all measurements with a resolution of 4 cm^−1^. Acquisition time of FTIR spectra for each sample is 5 min.

### 2.4. Nuclear Magnetic Resonance (NMR)

NMR measurements were carried out on an Agilent Technologies DD2 600 MHz NMR spectrometer equipped with 5 mm HCN cryoprobe, and on Bruker Avance Neo 600 MHz NMR spectrometer equipped with 5 mm BBO probe. NMR samples were prepared by dissolving 0.1 g of the sample in 0.6 mL D_2_O. NMR chemical shifts are reported in δ (ppm) relative to TMS (*δ* 0 ppm). Identification was performed using the characteristic NMR resonances, assigned on the basis of their chemical shifts as well as with the use of a set of 2D experiments: heteronuclear single quantum coherence (^1^H-^13^C HSQC) and heteronuclear multiple bond correlation (^1^H-^13^C HMBC). 1D ^1^H NMR spectra were recorded using *zgesgp* standard pulse sequence. Spectral width was set to 11,904 Hz (19.85 ppm). A relaxation delay of 5.0 s was used. ^1^H NMR spectra were recorded using 65,536 points and 16 scans. ^13^C NMR data were acquired using *deptqgpsp* pulse sequence. Spectral width was set to 35,714 Hz (236.68 ppm). A relaxation delay of 2.0 s was used. ^13^C DEPTQ spectra were recorded using 32,768 points and 10,000 scans. 2D ^1^H-^13^C HMBC spectra were recorded using *hmbcetgpl3nd* pulse sequence. Spectral width for proton dimension was 7812 Hz (13.0 ppm) and for carbon dimension 30,169 Hz (220 ppm). A relaxation delay of 1.25 s was used. The spectrum was recorded using 2048 points in F2 and 256 increments in F1 dimension, and 144 scans.

### 2.5. Potentiometric and Polyelectrolyte Charge Titration

Charges present on the polymer were quantified by pH-potentiometric and polyelectrolyte titration in water. For potentiometric titrations, samples were titrated in a forward and backward manner starting from acidic to alkaline with 0.1 mol/L HCl and 0.1 mol/L NaOH as titrants. A two-burette auto-titration unit T70 (Mettler Toledo, Greifensee, Switzerland) added the titrants, and a glass pH electrode DG-111 SC (Mettler Toledo, Greifensee, Switzerland) was used to measure the pH of the solution continuously. Nitrogen gas purging enabled an inert atmosphere during the measurements. The ionic strength was set to 0.1 mol/L (adjusted by addition of 3 mol/L KCl). The detailed description of measurements can be found elsewhere [22,23]. Polyelectrolyte titrations were used to determine the amount of protonated, non-succinylated amine groups of PAA. The cationic groups were titrated with anionic polyethylene sulphonic acid (PES-Na) (titrant) with the concentration of 0.001 mol/L. After incremental addition of the titrant (0.1–0.25 mL), with an auto-titration unit DL 53 (Mettler Toledo, Greifensee, Switzerland), the equivalence point was determined with a phototrode DP5 (Mettler Toledo, Columbus, OH, USA) at a wavelength of 660 nm with addition of Toluidine blue as indicator (c = 0.1 mmol/L). The degree of substitution with monosodium succinate was calculated from the amount of cationic charge obtained from the polyelectrolyte titration. PAA mass recovery was calculated using Equation (1).
(1)$$PAA\;recovery\;\left(\%\right)=\;100\;\%\;\times \frac{m\left(product\right)-m\left(mono\;sodium\;succinate\;substituent\right)\;}{m\;\left(\mathrm{PAA}\;\mathrm{used}\;\mathrm{as}\;\mathrm{reactant}\right)}$$

### 2.6. XRD Measurements and Optical Microscopy

Polyallylamine [24] and succinylated derivatives were subjected to X-ray diffraction measurements with D2 Phaser system (XRD), (Bruker, Germany). The samples were measured on Si-plates from 5°–70°with increment of 0.03° at 30 kV. The results are shown in the Appendix A (Appendix A). The powders were also compared with respect to their macroscopic morphology by light microscopy (Appendix A).

### 2.7. Evaluation of Biological Efficacy

#### 2.7.1. Biocompatibility Testing

Biological reactivity and potential cytotoxic activity of PAA and PAA-xSA samples were evaluated on mouse fibroblasts (NCTC clone 929: CCL 1; L929; American Type Culture Collection, ATCC, Manassas, VA, USA) in vitro. The L929 cells were seeded onto 96-well microplates (Nunc, Naperville, IL, USA) at a density of 10,000 cells/well, which corresponds to 50,000 cells/mL and were incubated for 24 ± 1 h at 37 ± 1 °C and 5 ± 1% CO_2_ in humidified atmosphere to attach. The cytotoxicity was evaluated after 24 ± 1 h of exposure with the 3 (4,5 dimethylthiazol-2-yl)-2,5 diphenyltetrazolium bromide (MTT) assay in accordance with the International Standard ISO 10993-5:2009, Biological evaluation of medical devices—Part 5: Tests for in vitro cytotoxicity. The biological reactivity was visually evaluated following exposure to the samples for 24 ± 1 and 48 ± 1 h, by light microscopy, and graded on a scale of 0 to 4 as described in ISO 10993-5:2009. The samples were dissolved in complete growth medium (MEM supplemented with 10% FBS, 4 mM L-glutamine, 0.11 mg/mL sodium-pyruvate, 100 IU/mL penicillin, and 100 µg/mL streptomycin) at the concentrations 2 mg/mL for PAA, 2.5 mg/mL for PAA-0.5SA, and 20 mg/mL for PAA-1SA, PAA-2.5SA, PAA-5SA, and PAA-10SA. Graded concentrations in the range of 0.001 to 20 mg/mL of PAA and PAA-xSA samples were tested, and half maximal inhibitory concentration (IC50) values determined. In each experiment, a negative control (complete growth medium) and a positive control (Etoposide 100 µL/mL) were included. Three independent experiments were performed in five replicates. The program GraphPad Prism 9 (GraphPad Software, San Diego, CA, USA) was used for statistical analysis and calculation of half maximal inhibitory concentration (IC50) values. ANOVA (one-way analysis of variance) and Dunnet’s multiple comparison test were used to determine statistically significant differences in cell viability between the control and the samples. The IC50 represents the concentration of a compound required for a 50% inhibition of growth in vitro. For PAA and PAA-xSA samples, the IC50 values were determined using nonlinear regression (log(agonist) vs. normalized response—variable slope) using GraphPad Prism 9.

#### 2.7.2. Antimicrobial Testing

For antimicrobial tests, the samples were dissolved in sterilized distilled water. From these solutions, the materials were subsequently diluted from base concentration of 400 mg/mL with two-fold dilution until 0.39 mg/mL. Final concentration in the test tube ranged from 20 mg/mL until 0.02 mg/mL. Two samples for each concentration were prepared. Antimicrobial activity was tested against *S. aureus* (DSM 799) and *P. aeruginosa* (DSM 1128). In the solutions, bacteria were added, and incubated for 24 h at 37 °C, where the growth of the bacteria was observed. In the case that the growth was not exactly determined, the material was applied on plates for probable determination of minimum inhibition concentration (MIC) and minimum bactericidal concentration (MBC).

### 2.8. Electrospinning of Fibers

#### 2.8.1. Preparation of Solutions

An aqueous solution of 10% (*w*/*v*) was prepared by dissolution of 10 g of PVA in 100 mL of Ultra-pure water and heated at 90 °C for 3 h with stirring (200 rpm) of the solution until homogeneity was achieved. This solution was used as a base for preparation of the following material for electrospinning. Into the base solution of PVA, PAA, or its succinylated products were added in different weight ratio (10%, 15%, and 20% (*w*/*w*)) with respect to the mass of PVA in the solution. To obtain completely uniform mixtures, after the addition of the synthesized material, it was vigorously mixed (200 rpm) at 25 °C to avoid thermal degradation of the amides added in the solution. For each solution, viscosity (Fungi lab rheometer, Hauppauge, New York, NY, USA), and conductivity (Mettler Toledo, Greifensee, Switzerland) was measured. Fifteen solutions were prepared, labelled as (PAA-xSA_y) with x as the molar ratio of SA with respect to the PAA amine during the derivatization, and y as the weight percent of PAA-xSA added to the spinning solution of 100 mL with respect to the mass of PVA.

#### 2.8.2. Production and Characterization of Fibers

For production of electrospun fibers, a home-built single-needle electrospinning setting was used comprising a syringe pump NE4000 (New Era Pump System, Inc., New York, NY, USA) and a CZE1000R (Spellman High Voltage Electronics Corp., Bochum, Germany) providing high voltage power supply. The polymer solution was pushed through a 5 mL plastic syringe equipped with a 90° blunt stainless-steel needle of 0.9 × 40 mm (20 G) dimensions. As collector an aluminum foil wrapped around 250 × 250 mm aluminum plate was used. Production of fibers was accomplished at parameters confirmed for the production of PVA fibers, including: flow rate 0.45 mL/h, 20 cm distance from the tip of the needle to the collector, voltage 25 kV with negative polarity. Temperature (20 °C) and rel. humidity (27%) of the surroundings were kept constant. Collected (ca for 10 min) fibers were characterized using Keyence VHX 7000 optical microscope (Osaka, Japan) and Field Emission Scanning Electron Microscopy by using a Carl Zeiss FE-SEM SUPRA 35 VP electron microscope. All samples were placed on aluminum holders using double-side carbon conductive tape. A thin layer of palladium was sputtered over the samples using a Benchtop Turbo sputtering device (Dentum Vacuum, Moorestown, NJ USA). An accelerating voltage of 1 keV and 4.5 mm was used for recording the SEM images. The magnification used for providing SEM images ranged from 500-fold for 20 µm to 60,000-fold for 200 nm. The diameters of the electrospun fibers were measured directly from selected SEM images using Gwyddion (a multiplatform modular free software, Czech Metrology Institute, Brno, Czech Republic) and are given as an average value with the standard error for each sample, calculated from at least 200 measurements in one image, with three repetitions per polymer type.

## 3. Results and Discussion

### 3.1. ATR-IR

Figure 2 shows the ATR-IR spectra of PAA and the succinylated polymers. PAA (c) shows typical peaks for the primary amine (NH, 3400 cm^−1^) and alkyl (CH, 2906 cm^−1^) groups [24]. The carboxyl hydroxyls (OH, 3260 cm^−1^; 1400 cm^−1^) and alkyls (CH 2930 cm^−1^) of the succinate substituent increased with higher amounts of anhydride used during the reaction. Similarly, the carbonyls of the amide (C=O 1640 cm^−1^) and the NH vibration of the amide and amines (NH 1560 cm^−1^) are visible. Both bands can also contain the carbonyl Comparing the protons and carbon vibration of the carboxylate sodium salt.

### 3.2. NMR

The carbon signal of the amide bond for the sample PAA-1SA in D_2_O appears at 177.78 ppm in the ^13^C NMR spectra (Figure 3d) compared to the succinate carbonyl at 179.69 ppm. Peaks in the ^1^H NMR correspond to protons of the succinate (2.67 and 2.56 ppm Figure 3c). For PAA-1SA, no full amidation could be observed. Broader peaks in the ^1^H NMR at 3.12, 1.74, and 1.29 ppm are corresponding to the protons of unreacted PAA.

With evident shifts of protons and carbons of PAA, it can be concluded that there is a formation of the amide bond. This can be explained with the resonance structures of the amide causing magnetic anisotropy of the amide. Due to its unstable nature regarding charge division, it can cause shielding or deshielding of nearby protons. Comparing the protons and carbon of the methylene groups further from the amine and amide group, there is a slight decrease of the chemical shifts which is caused by shielding behavior of the protons due to their resonance structure [25]. This behavior can be observed at 1.74 and 1.96 ppm (Figure 3a) and 45.80 and 37.04 ppm (Figure 3d), which both correspond to the methylene groups of PAA, and the latter being attributed to the carbon of the secondary amide.

Peaks in the ^1^H-^13^C HMBC spectra (Figure 4) mostly show the correlation between atoms of the succinate residue, which in combination with provided ^1^H NMR and ^13^C NMR spectra support that there is a formation of a covalent bond between the PAA and succinic acid. PAA: ^1^H NMR (600 MHz, D_2_O, δ in ppm TMS) δ 3.10, 2.09, 1.95, 1.56. PAA: ^13^C NMR DEPTQ (151 MHz, D_2_O, δ in ppm TMS) δ 45.45, 45.16, 36.39, 32.67. PAA-1SA: ^1^H NMR (600 MHz, D_2_O, δ in ppm TMS) δ 3.12, 2.67, 2.56, 1.96, 1.74, 1.29. PAA-1SA: ^13^C NMR DEPTQ (151 MHz, D_2_O, δ in ppm TMS) δ 179.69, 177.78, 45.80, 37.42, 35.04, 33.29, 32.31, 31.70.

### 3.3. Titration, DS, and PAA Recovery

The measured p*K*a of the primary amine groups in PAA is 8.3 [26] (Figure 5a). When increasing the DS from 0.2 (PAA-0.5SA) to 1 (PAA-10SA) the p*K*a of the remaining primary amino groups increased from 9.4 to over 11. This was an indirect indication of a successful succinylation of PAA via amide formation and for the presence of succinate residues. Substitution obviously led to the fact that the remaining amine groups were more alkaline compared to free PAA. The p*K*a values for the amine were not detectable for samples PAA-5SA and PAA-10SA due to the almost full derivatization.

Free carboxyl groups of succinic acid (3.5 < pKa < 5.6) were detected in all derivatized PAA samples by potentiometric charge titration.

The amount of primary amino groups in PAA and all PAA-xSA samples was also determined by polyelectrolyte titration. As seen in Figure 5b, calculated DS values correlate with the charge ratios obtained from potentiometric titration. A variety of ampholytic polymers were obtained by succinylation. It is, however, not possible to determine the substitution pattern along the polymer chains, or the substitution in a set of distributed chain lengths. Unmodified PAA has a cationic charge of 9.26 mmol/g. This is reduced to 8.84 mmol/g for PAA-0.5SA. For PAA-5SA and PAA-10SA, no cationic charge could be observed, indicating complete succinylation of all primary amino groups. Though the reaction was conducted in water, a sufficient DS can still be reached by the addition of a suitable amount of succinic anhydride. The amount of SA allowed a control over the charge ratios in ampholytic polyelectrolytes. PAA recovery rates were from 56 wt.% for PAA-0.5SA, 45 wt.% for PAA-1SA, 46 wt.% for PAA-2.5SA, 37 wt.% for PAA-5SA, and 53 wt.% for PAA-10SA and could potentially be improved by alternative purification procedures such as ultrafiltration [27].

### 3.4. Biological Activity of PAA and the PAA-xSA Samples

#### 3.4.1. Biocompatibility Testing

Succinylation of PAA significantly influenced biocompatibility. Changes in cell viability (Figure 6a) and IC_50_ values (Figure 6b) for PAA and the succinylated products are given for L929 mouse fibroblasts. Unmodified PAA completely inhibited growth of L929 cells already at a concentration of 0.2 mg/mL. With the reduction of the positive charges, the cytotoxic activity decreased and the IC_50_ gradually increased in line with the degree of succinylation, reaching a maximum of 7.0 ± 2.2 mg/mL for PAA-5SA. However, at higher degrees of succinylation (PAA-10SA) a lower IC_50_ was observed. This can be attributed either to the presence of minor residues of free SA, or the fact that the high negative charge density can also cause intracellular damage, as observed for anionic nanoparticles [28]. Biological reactivity evaluated under the light microscope reflected the findings of the cytotoxicity assay. While native PAA caused severe reactivity (reactivity grade 4), nearly destroying the cell culture already at 0.2 mg/mL, the PAA-xSA samples showed decreased biological reactivity (Figure 7). All PAA-xSA samples caused no (grade 0), slight (grade 1), or mild (grade 2) reactivity at lower concentrations (0.2–2.5 mg/mL), inducing significant morphological changes that were graded as moderate (grade 3) and severe reactivity (grade 4), only at higher concentrations (5–20 mg/mL). The lowest biological reactivity was observed after exposure to PAA-5SA. Interestingly at the highest tested concentration (20 mg/mL) PAA-1SA was the least reactive, at both time points.

Overall cytotoxicity of the PAA-xSA samples is still high and the calculated IC50 values are still in the range of published values for quaternized PAA [6]. The polymers could however still be used for, e.g., drug delivery studies, measuring cell uptake at significantly lower concentrations [29,30].

#### 3.4.2. Antimicrobial Testing

Succinylation also decreased the antimicrobial activity. Native PAA showed complete inhibition of the bacterial growth for *S. aureus* at a concentration of 0.08 mg/mL (Table 1), and *P. aeruginosa* partial inhibition at 0.04 mg/mL (Appendix A). However, already with minor derivatization of the amine group there was a very significant increase of the minimum inhibitory concentration (MIC). A substitution of 0.7 (PAA-1SA) has a MIC above 20 mg/mL. Our results show that PAA succinylation decreased its deleterious effects in both test systems increasing survival in the tested bacteria species as well as in L929 mouse fibroblast and suggest the latter may be more sensitive to the presence of residual amino groups in derivatized PAA.

Comparing the antimicrobial activity of PAA-5SA and PAA-10SA samples at the final concentration of 10 and 20 mg/mL, there is a slight antimicrobial activity against *S. aureus* (Appendix A), and no antimicrobial properties against *P. aeruginosa* (Appendix A). This can be attributed to the high concentration of added succinic anhydride.

### 3.5. Production of PVA/PAA-xSA Fibers

Obtained electrospun fibers were characterized regarding their structure and fiber diameter. PAA is not electro-spinnable but in combination with PVA. 10% (*w*/*v*) of PVA showed good characteristics regarding conductivity and viscosity for electrospinning. Conductivity of the solutions ranged from 2400 µS/cm to 2700 µS/cm and the viscosity ranged from 206 to 280 mPa s depending on the PAA-xSA material and the mass added to the PVA solution. Conductivity and viscosity values are similar as the ones reported for PVA/chitosan electrospinning solutions [31]. The spinning parameters were adjusted for PVA, and the same conditions were applied for PAA and its derivatives. A DC voltage of 25 kV and a distance of 20 cm from the collector showed good spinnability of the fibers (Figure 8a).

According to the SEM images, average fiber diameter was around 100–250 nm (Appendix A). With the increase of the added amount (10–20% (*w*/*w*)) of the PAA and PAA-xSA compounds, there was an increase of the obtained fiber diameters and decrease of formation of beads (Figure 8 and Appendix A). Comparing the diameter of fibers of PAA-xSA, there was a constant increase of the diameter of fibers going from PAA-0.5SA (125 nm) < PAA-1SA (173 nm) < PAA-2.5SA (266 nm) with the increase of molar ratio of added SA after added 10% (*w*/*w*) (Appendix A). The exception was PVA/PAA-5SA fibers in which there was a decrease to 235 nm (PAA-5SA_10) to 119 nm for added 10% (*w*/*w*)) of PAA-10SA, and even bigger reduction of fiber diameter with the increase of added amount of PAA-SA compound from 117 nm to 208 nm for PAA-0.5SA to PAA-10SA for added 15% (*w*/*w*) into PVA solution (Appendix A). Similar behavior can be seen for added 20% (*w*/*w*) of PAA-xSA material. The behavior can be explained with the excess of unbounded SA, which could also have caused increase of the cytotoxicity of the material.

## 4. Conclusions

Polyallylamine hydrochloride (PAA) could be succinylated in alkaline water with different degrees of substitution by a straightforward reaction with succinic anhydride. The succinylation of the primary amines forming amides could be confirmed by ATR-IR, ^1^H, ^13^C DEPTQ, and 2D HMBC NMR measurements. With an increase in the molar excess of succinic anhydride in the reaction, a variety of polyampholytes were obtained comprising primary amines and sodium carboxylates along the polymer chains. Polyelectrolyte titration methods correlated well with potentiometric charge titrations and allowed for the determination of the degree of succinylation (DS). A reduction of the amount of cationic charge by the succinylation, led to a change in the protonation behavior of the residual amino groups. A certain buffering capacity in the vicinity of neutral pH values was observed, showing some similarities to polyampholytic amino acid side chains.

Gradual substitution of amine groups caused an increase of biocompatibility observed in the L929 mouse fibroblast model system. The half maximal inhibitory concentration (IC_50_) of the derivatized polymer in mouse fibroblasts could be increased with the degree of succinylation and was 7.0 ± 2.2 mg/mL for a polymer with DS of 1. An abrupt reduction of antimicrobial properties was observed at a DS of 0.7. At a high excess of added succinic anhydride, removal of unbounded succinic acid has its limitations, most likely causing certain antimicrobial effects against Gram negative bacteria.

In combination with uncharged polyvinyl alcohol (PVA), PAA and succinylated derivatives produced homogeneous electrospun nano-fibers, with an average fiber diameter ranging from 100 to 250 nm. Though spinnability and production of PVA/PAA-SA fiber webs is possible at acceptable yields, an application as wound dressing might be limited by the relatively high cytotoxicity even after succinylation. Future research on the biological effects of similar polyampholytes should include detailed mechanistic studies of cell uptake and membrane interaction.

## Figures and Tables

**Figure 1 polymers-13-02840-f001:**
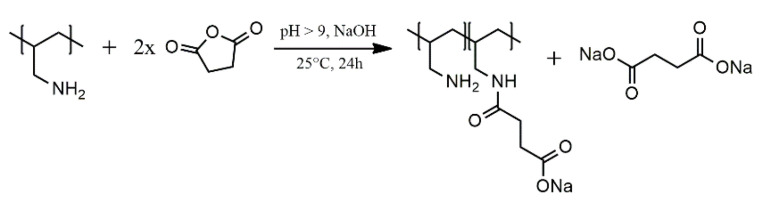
Reaction scheme for the succinylation of PAA and the hydrolysis of succinic anhydride.

**Figure 2 polymers-13-02840-f002:**
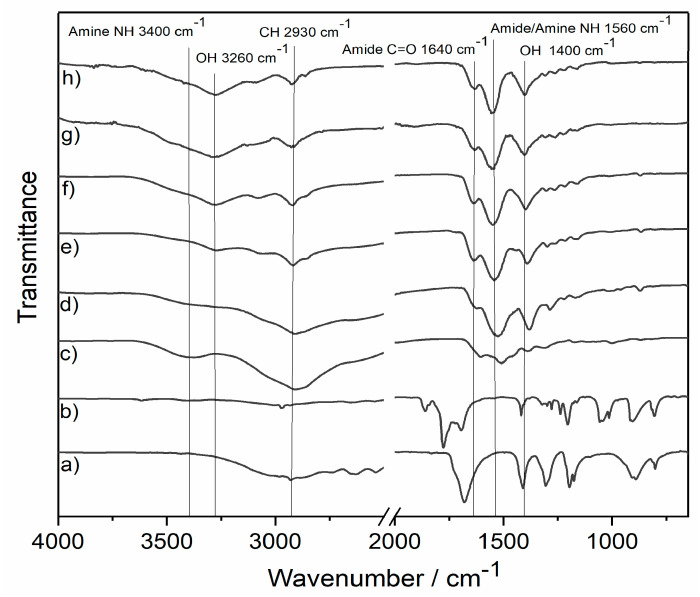
ATR-FTIR spectra of (**a**) succinic acid, (**b**) succinic anhydride, (**c**) PAA, (**d**) PAA−0.5SA, (**e**) PAA−1SA, (**f**) PAA−2.5SA, (**g**) PAA−5SA, and (**h**) PAA−10SA.

**Figure 3 polymers-13-02840-f003:**
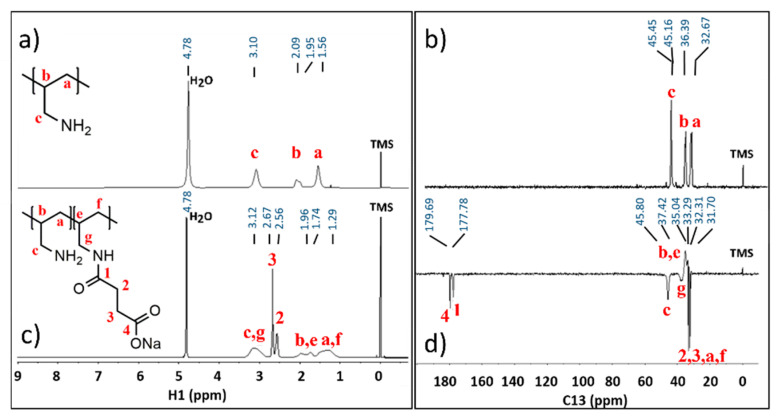
(**a**) ^1^H and (**b**) ^13^C of PAA; (**c**) ^1^H and (**d**) ^13^C DEPTQ NMR spectra of PAA-1SA in D_2_O.

**Figure 4 polymers-13-02840-f004:**
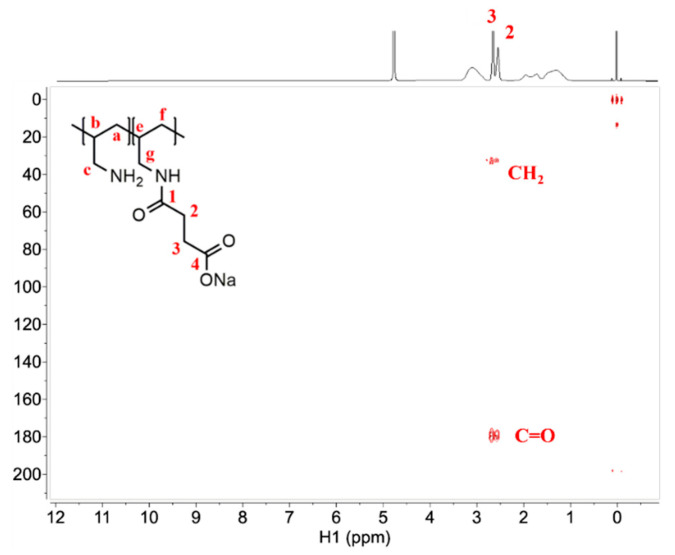
^1^H-^13^C NMR HMBC spectra of PAA-1SA.

**Figure 5 polymers-13-02840-f005:**
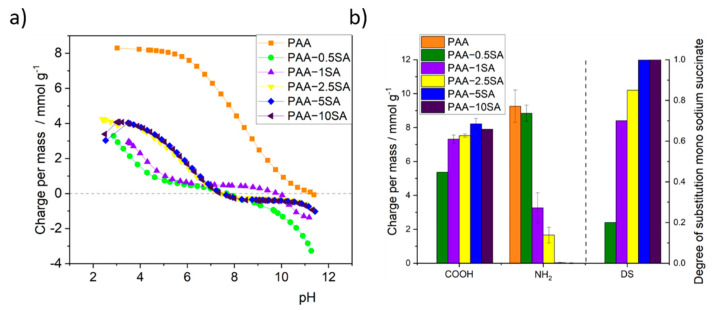
(**a**) Potentiometric titration isotherms and (**b**) charge per mass and degree of substitution determined by polyelectrolyte titration of PAA and PAA-xSA.

**Figure 6 polymers-13-02840-f006:**
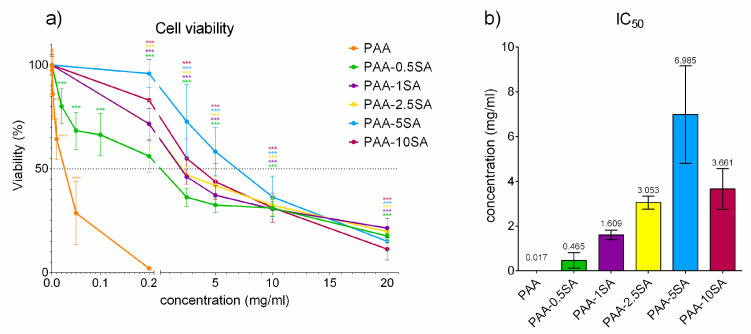
Cytotoxicity test of PAA and PAA-xSA samples. (**a**) Changes in cell viability after 24 h of exposure of L929 cells to PAA and PAA-xSA samples. Data are presented as percentage of the negative control (complete growth media). (**b**) IC_50_ values for PAA and PAA-xSA samples in the L929 cell line. The asterisks (*) denote statistically significant difference (ANOVA and Dunnet’s multiple comparison test) between the control and exposed cells (** *p* ≤ 0.01; *** *p* ≤ 0.001).

**Figure 7 polymers-13-02840-f007:**
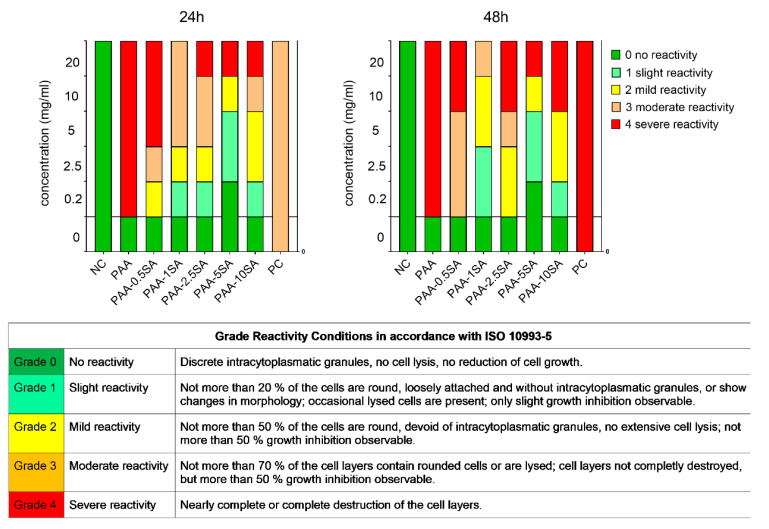
Biological reactivity of PAA and PAA-xSA samples (**left**) after 24 h of exposure; (**right**) after 48 h of exposure. Morphological changes were evaluated under the microscope and graded according to ISO 10993-5. NC is the negative control (complete growth medium) and PC is the positive control (Etoposide 100 µg/mL). Both controls were tested at only one concentration and a uniform bar is shown for visualization.

**Figure 8 polymers-13-02840-f008:**
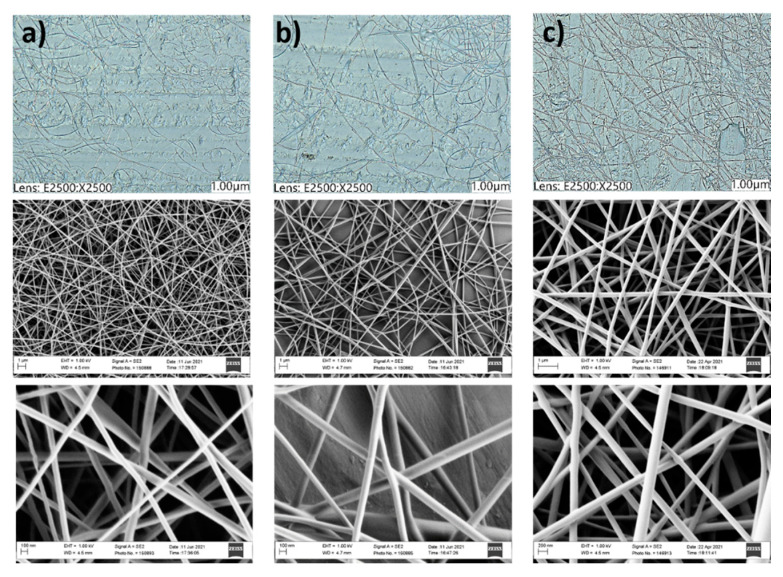
Optical microscope (above) and SEM images of (**a**) PVA, (**b**) PAA-1SA_10, (**c**) PAA-10SA_10.

**Table 1 polymers-13-02840-t001:** MIC and MBC for PAA and PAA-xSA.

	S. Aureus		P. Aeruginosa	
	MIC	MBC	MIC	MBC
PAA	0.08 mg/mL	0.16 mg/mL	0.08 mg/mL	0.16 mg/mL
PAA-0.5SA	Insoluble in culture	Insoluble in culture	Insoluble in culture	Insoluble in culture
PAA-1SA	>20 mg/mL	n.d.	>20 mg/mL	n.d.
PAA-2.5SA	>20 mg/mL	n.d.	>20 mg/mL	n.d.
PAA-5SA	>20 mg/mL	n.d.	>20 mg/mL	n.d.
PAA-10SA	>20 mg/mL	n.d.	>20 mg/mL	n.d.

## Data Availability

The original contribution for this study is included in the article.

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
