# Peer review of "Succinylation of Polyallylamine: Influence on Biological Efficacy and the Formation of Electrospun Fibers"

_polymers, 2021, doi:10.3390/polym13172840_

Round 1
Reviewer 1 Report
The reviewed manuscript presents extensive comparative studies including various amounts of succinic anhydride Indeed, by the authors, such tests are extensively described in the literature data for new products, while the polyallylamine derivatives was analysed with respect to molecular structure through nuclear magnetic resonance and infra- red spectroscopy provides interesting data. For this reason, I find the article thematically suitable for Polymers. Also, because the authors included electrospinning of succinylated PAA to their studies, the manuscript sufficiently provide biocompatibility in the mouse fibroblast model system . However, there are few issues that should be resolved before potential acceptance.
1) The authors can do a better bibliographic research. References are not update. Please update and include 5 recent years' related references.
2) Add more reference in discussion part in order to support the statement.
3) Conclusion is not clear, should be rewritten caring order, and clarity of exposition.
Author Response
Replies to comments of reviewer 1
We would like to thank the reviewer for the useful suggestions to improve the quality of our manuscript. Our answers to the comments can be found below. They also refer to the changes made by indicating the line numbers in the submitted revision R1 in which changes were highlighted in yellow.
Comments and Suggestions for Authors
The reviewed manuscript presents extensive comparative studies including various amounts of succinic anhydride Indeed, by the authors, such tests are extensively described in the literature data for new products, while the polyallylamine derivatives was analysed with respect to molecular structure through nuclear magnetic resonance and infra- red spectroscopy provides interesting data. For this reason, I find the article thematically suitable for Polymers. Also, because the authors included electrospinning of succinylated PAA to their studies, the manuscript sufficiently provide biocompatibility in the mouse fibroblast model system . However, there are few issues that should be resolved before potential acceptance.
- The authors can do a better bibliographic research. References are not update. Please update and include 5 recent years' related references.
Answer: We thank reviewer 1 for this comment. We agree that the references could be updated with more actual work. If we understand correctly, references should be updated with publications from the past 5 years. We have therefore thoroughly revised the reference list citing very recent literature in the introduction of the manuscript. The updated references are 1-21 in the lines 41-74 of the introduction.
- Add more reference in discussion part in order to support the statement.
Answer: We agree that proper referencing can better support the statements in the discussion part. We have therefore added the following references (given at the end of the revised manuscript and indicated with the line number here and in the revised version) at each statement as follows:
Line 242: “Figure 2 shows the ATR-IR spectra of PAA and the succinylated polymers. PAA (c) shows typical peaks for the primary amine (NH, 3400 cm-1) and alkyl (CH, 2906 cm-1) groups [24].”
Line 266: “Comparing the protons and carbon of the methylene groups further from the amine and amide group there is a slight decrease of the chemical shifts which is caused by shielding behavior of the protons due to their resonance structure [25].”
Line 307: PAA recovery rates were from 56% wt.% for PAA-0.5SA, 45 wt.% for PAA-1SA, 46 wt.% for PAA-2.5SA, 37 wt.% for PAA-5SA and 53 wt.% for PAA-10SA and could potentially be improved by alternative purification procedures such as ultrafiltration [27]
Line 320: This can be attributed either to the presence of minor residues of free SA, or the fact that the high negative charge density can also cause intracellular damage, as observed for anionic nanoparticles [28].
Line 332: Overall cytotoxicity of the PAA-xSA samples is still high and the calculated IC50 values are still in the range of published values for quaternized PAA [29]. The polymers could however still be used for e.g. drug delivery studies measuring cell uptake at significantly lower concentrations [30],[31].
Line 372: Conductivity and viscosity values are similar as the ones reported for PVA/chitosan electrospinning solutions[32]
3) Conclusion is not clear, should be rewritten caring order, and clarity of exposition.
Answer: Thank you for the comment. We have completely rewritten the conclusion and ordered it chronologically following the same systematics as the results part. We have also added a suggestion on potential applications and further studies related to the type of polyampholytes investigated in this study. We hope that the conclusion is now acceptable from the reviewer's point of view (line 395).
Submission Date
23 July 2021
Date of this review
02 Aug 2021 08:08:05
Reviewer 2 Report
This manuscript investigates the succinylation of primary amines of the synthetic polyelectrolyte polyallylamine and its influence on the cytotoxicity and antibacterial activity of the resulting polymers, and the possibility to produce electrospun fibers. The evaluations of results are sufficient, clear, and contain sufficient information and most of the Figures are also clear and demonstrative. However, a few issues need to be solved:
1- 20 analyses to check the fiber average are not enough. Usually, the authors report an average of 200 fiber diameter size. Please increase the number of analyses.
2- The Figure related to FTIR is not completely clear cause the bands marked don't correspond with the one written in the test.
3- there is a mistake with the label of paragraph 2.4 ( Biological activity of PAA and the PAA-xSA samples). It should be 3.4, etc. Please fix it.
4- No paragraph related to statistical analysis is indicated in the manuscript. No statistical analysis has been performed. without that, data are shown in Figures 5b and 6a, 6b, and Table 3 have no significance. Please provide it.
5- Some parameters about analysis are missing like which magnification has been used to provide SEM images? The acquisition time for FTIR spectra? The concentration of cells used (cells/well or cells/ml)?
Author Response
Comments and Suggestions for Authors
This manuscript investigates the succinylation of primary amines of the synthetic polyelectrolyte polyallylamine and its influence on the cytotoxicity and antibacterial activity of the resulting polymers, and the possibility to produce electrospun fibers. The evaluations of results are sufficient, clear, and contain sufficient information and most of the Figures are also clear and demonstrative. However, a few issues need to be solved:
1- 20 analyses to check the fiber average are not enough. Usually, the authors report an average of 200 fiber diameter size. Please increase the number of analyses.
Answer: We thank the reviewer for this comment and agree that better statistics are obtained from a larger number of samples. We have now performed at least 200 measurements in the image analysis from three repetitions per sample and calculated the average fiber diameter. Histograms of the fiber distribution (Figure S3) were added to the supporting information. We have amendend the experimental part explaining this (line 229- line 234)
2- The Figure related to FTIR is not completely clear cause the bands marked don't correspond with the one written in the test.
Answer: We thank the reviewer for this comment and have to apologize for this mistake. We have now added the wavenumbers for essential peaks in Figure 2 and reformatted it, and also synchronized the numbers with the text in line 243 - 248.
3- there is a mistake with the label of paragraph 2.4 ( Biological activity of PAA and the PAA-xSA samples). It should be 3.4, etc. Please fix it.
Answer: We have made the changes suggested (line 312).
4- No paragraph related to statistical analysis is indicated in the manuscript. No statistical analysis has been performed. without that, data are shown in Figures 5b and 6a, 6b, and Table 3 have no significance. Please provide it.
Answer: We thank the reviewer for the remark. The statistical analysis was now performed (one-way ANOVA) and the paragraph explaining the statistical analysis was included in the Section “2.7.1 Biocompatibility testing« (line 181 - 188) and is as follows: » The program GraphPad Prism 9 (GraphPad Software, San Diego, CA, USA) was used for statistical analysis and calculation of half maximal inhibitory concentration (IC50) values. ANOVA (one-way analysis of variance) and Dunnet’s multiple comparison test were used to determine statistically significant differences in cell viability between the control and the samples. The IC50 represents the concentration of a compound required for a 50 % inhibition of growth in vitro. For PAA and PAA-xSA samples, the IC50 values were determined using nonlinear regression (log(agonist) vs. normalised response—variable slope) using GraphPad Prism 9.” Correspondingly, the Figure 6a was corrected and the information on statistical analysis was added to the Legend of the Figure 6.
5- Some parameters about analysis are missing like which magnification has been used to provide SEM images? The acquisition time for FTIR spectra? The concentration of cells used (cells/well or cells/ml)?
Answer: We thank the reviewer for the remark. The SEM magnification was between 0.5 k – 60 k fold and was added to line 229-234. The acquisition time for FTIR spectra was 5 min and added to line 115. The cell density was 10 000 cells/well, which corresponds to 50 000 cells/mL. This information was added to line 163.
Submission Date
23 July 2021
Date of this review
06 Aug 2021 12:28:55
Additional changes made during the revision by the authors:
We have slightly reformatted the chemical structures and captions in Figures 1, 3, 4 to increase legibility without changing the content and main findings. We have added a list of the ppm shifts of NMR peaks in line 275 – 279. We have included an additional funding source for the electron microscope used in the study.